# Two-Photon and Multiphoton Microscopy in Anterior Segment Diseases of the Eye

**DOI:** 10.3390/ijms25031670

**Published:** 2024-01-30

**Authors:** Merrelynn Hong, Shu Zhen Chong, Yun Yao Goh, Louis Tong

**Affiliations:** 1Yong Loo Lin School of Medicine, National University of Singapore, Singapore 117597, Singapore; merrelynn@gmail.com; 2Training and Education Department, Singapore National Eye Centre, Singapore 168751, Singapore; 3Singapore Immunology Network (SIgN), Agency for Science Technology and Research (A*STAR), Singapore 138632, Singapore; chong_shu_zhen@immunol.a-star.edu.sg; 4Lee Kong Chian School of Medicine, National Technical University, Singapore 639798, Singapore; yunyaogoh@gmail.com; 5Corneal and External Diseases Department, Singapore National Eye Centre, Singapore 168751, Singapore; 6Ocular Surface Group, Singapore Eye Research Institute, Singapore 169856, Singapore; 7Eye Academic Clinical Program, Duke-NUS Medical School, Singapore 169857, Singapore

**Keywords:** ocular surface, cornea, dry eye disease, two-photon microscopy, multi-photon microscopy, ocular imaging, animal models

## Abstract

Two-photon excitation microscopy (TPM) and multiphoton fluorescence microscopy (MPM) are advanced forms of intravital high-resolution functional microscopy techniques that allow for the imaging of dynamic molecular processes and resolve features of the biological tissues of interest. Due to the cornea’s optical properties and the uniquely accessible position of the globe, it is possible to image cells and tissues longitudinally to investigate ocular surface physiology and disease. MPM can also be used for the in vitro investigation of biological processes and drug kinetics in ocular tissues. In corneal immunology, performed via the use of TPM, cells thought to be intraepithelial dendritic cells are found to resemble tissue-resident memory T cells, and reporter mice with labeled plasmacytoid dendritic cells are imaged to understand the protective antiviral defenses of the eye. In mice with limbal progenitor cells labeled by reporters, the kinetics and localization of corneal epithelial replenishment are evaluated to advance stem cell biology. In studies of the conjunctiva and sclera, the use of such imaging together with second harmonic generation allows for the delineation of matrix wound healing, especially following glaucoma surgery. In conclusion, these imaging models play a pivotal role in the progress of ocular surface science and translational research.

## 1. Introduction

Two-photon excitation microscopy (TPM) and multiphoton fluorescence microscopy (MPM) are advanced forms of non-invasive, intravital high-resolution functional microscopy techniques that image dynamic molecular processes and resolve the features of the biological tissues of interest [1]. The advantages of using these methods in research include being minimally invasive for organs that are accessible [2] and allowing for deep levels of tissue penetration and high resolution [1]. In addition, TPM and MPM allow researchers to collect information from a single tissue sample with multiple colors of fluorescence, hence allowing the analysis of up to three parameters simultaneously from the same biologically active tissue. Because of this, powerful analyses involving the relationship of different proteins or other molecules can be investigated in a correlative way. Very often, this allows for longitudinal observations, including experiments with interventions, and may reduce the need for ex vivo tissue or the number of experimental animals [3].

Before the era of sophisticated in vivo imaging, most of the progress in knowledge concerning biological physiology and biochemistry relied on the use of cultured cells and tissues. However, such experimental studies usually fail to recapitulate the complex behavior of cells in their native tissues and organisms, which is largely due to systemic influences. Studies such as those in immunology involve the circulation of cells through many organs and extrapolation from cultured cells is of limited utility. Therefore, high-impact studies now require experimental animals, and effective imaging of such animals is highly advantageous in their evaluation. In addition, the ability of TPM to achieve a sub-micron resolution of imaged eye structures gives TPM a distinct advantage over other classical imaging modalities like optical coherence tomography (OCT) or confocal microscopy.

## 2. Applied Physics

The optics of such microscopy techniques have been described recently (Figure 1). MPM utilizes the combination of two or more photons to generate high-resolution, 3-dimensional images via the use of confocal microscopy [4]. This comprises two-photon excitation fluorescence microscopy (TPM or TPEF), in which two photons are absorbed by a target and its fluorescence is measured, and second harmonic generation (SHG), in which two photons interact concurrently with a material lacking inversion symmetry and combine to produce a photon with twice the energy of the incident photons in a scattering signal. The combination of both the emitting and scattering signals provides useful insight into biological tissues and molecular processes [4]. The modern rendition of the MPM technique relies upon fluorophores, substances which produce a new wavelength of light when stimulated with photons, concurrently absorbing two infrared photons. This results in spatially localized excitation and the production of high-resolution (0.4 microns) fluorescence images hundreds of microns into tissues. The use of infrared light also minimizes toxicity to mammalian cells and is suitable for repeated imaging [1].

In ophthalmology, molecules of interest include endogenous fluorophores such as vitamin A derivatives, which are excited by short-wavelength light (<400 nm) such as blue and ultraviolet light. These shorter-wavelength lights are, however, taken up and dispersed by the anterior segment of the eye [5]. The ultraviolet spectrum’s light damages living tissues by causing photochemical damage to DNA, inducing redox stress and leading to irrevocable opacification of the human crystalline lens and cornea [6].

TPM circumvents the above problem by utilizing longer-wavelength light, such as that of infrared, and nonlinear excitation to avoid absorption by chromophores for ultraviolet and visible light, allowing the use of reduced laser power to achieve subcellular-resolution imaging [2] with little to no tissue damage [1,7,8,9]. The use of near-infrared (NIR) light allows deep penetration into the retina and the excitation of retinal fluorophores, allowing the monitoring of metabolic transformation of visually significant molecules such as retinoid intermediates, i.e., retinyl esters and retinol [2], which are significant for the regeneration of chromophores [10]. Condensation products such as N-retinylidene-N-retinylethanolamine (A2E), precursors of toxic retinoids which are highly fluorescent, also act as biomarkers of diseases such as age-related macular disease and Stargardts [11].

A similar technique using fluorescence detection is flat fluorescence microscopy, which utilizes a thin mask placed between the sample and the image sensor. The input is encoded when passing through the mask until it reaches the image sensor, and thereafter a computational reconstruction of the image is performed [12].

Another similar technique that allows for deeper imaging of biological tissues is three-photon microscopy, in which three photons simulate a fluorophore and excite it to a higher state of energy. This, however, comes with increased risk of tissue damage in view of the higher density of photons, as well as photodamage and photobleaching [13].

## 3. History of Use in the Eye

TPM has been used recently in retina and other parts of the eye (Figure 2). For instance, a study used tetracycline-induced fluorescence to examine small molecules measuring less than 10 microns across, namely hydroxyapatite and whitlockite, which are sub-RPE molecules that are postulated to be precursors of drusen formation and hence represent potential for AMD [14], and which would otherwise not be picked up using conventional modalities. TPM similarly allows for the examination of microscopic structures, as demonstrated in a study where the interaction between microglial and macrophages in zebrafish retina was examined, revealing that post-injury dexamethasone inhibited microglial reactivity in response to cell death, as well as decreasing systemic dexamethasone toxicity and increasing stem cell proliferation rates to improve regeneration effects [15]. Other applications of this include the examination and quantification of the leukocyte infiltration of mouse retina [16], and in vivo imaging of crystalline lens, including cellular organization and in vivo environmental and structural support [17], which would otherwise be impossible to accomplish without TPM.

The setup of an MPM microscope first requires the use of a custom-built microscope, which is cheaper but requires technical expertise, or the conversion from a confocal microscope, which is more costly in view of the major modifications required [18]. Secondly a pulsed laser source is needed. Currently, femtosecond laser systems with near-infrared emissions can be acquired commercially, the most common of which is the titanium–sapphire (Ti:sapphire) oscillator, which has a wide range of wavelengths from 670 to 1070 nm, suitable for the study of a wide range of fluorophores in microbiology [19]. Subsequently, an optimal pulsed length is determined. This is however empirical and varies by microscope and tissues of interest being studied [20,21]. The excitation pathway of an MPM starts from the pulsed laser, travelling through a telescope and thereby expanding the beam. This can be prior to or followed by the addition of a λ/2 waveplate and a polarizer, which serve as modulators of laser intensity. This allows for the easy adjustment of the laser intensity. This laser beam is then detected by a xy deflection module and expanded to fill up the back aperture of the objective, which focuses the light onto the specimen of interest. The resulting highly scattered photons reflected off the specimen are detected via a “whole-area” epi-detection scheme, in which all the reflected photons are directed through to a detector [18].

This narrative review aims to describe the contribution of two-photon microscopy to ocular surface and anterior segment research.

## 4. Methodology

Databases of Entrez and PubMed were searched from 1 January 2006 to 26 November 2023, using keywords “two photon microscopy” and “ophthalmology” OR “anterior segment” OR “ocular surface”. Some 461 results were found. Through the search, we selected and reviewed articles on the utilization of TPM in the study of the ocular surface or the anterior segment of the eye.

## 5. Corneal Immunology

The cornea is one of the rare tissues of the human body granted immune privilege. In the physiological state, the cornea is continually in contact with foreign antigens, pathogens, and allergens, yet this does not mount significant inflammation or immune responses. Previously, this immune privilege was thought to be due to the lack of resident corneal immune cells during normal conditions. However, present-day reports have shown that conventional dendritic cells (cDCs) and macrophages resides in the cornea as the resident corneal immune cells, even in the absence of disease [22,23].

Corneal infections may have disastrous sequelae. In developed countries, herpes simplex virus 1 (HSV-1) keratitis remains the leading cause of blindness [24]. During HSV-1 infections, CD4^+^ T cells are recruited into the cornea and subsequently induce inflammatory processes such as stromal keratitis [25], while CD8^+^ T cells that are recruited into the cornea suppress the viral load [26] and propagate the formation of lymphatic vessels.

Downie et al. developed a functional in vivo confocal laser microscopy system (label-free system unlike TPM or MPM) to track immune cells in different depths of the cornea in living humans. Using time lapse videos, the study found three types of immune cells in the cornea: stromal macrophages, stromal T cells, and intraepithelial dendritic cells [27]. The shape and behavior of these human intraepithelial dendritic cells resembled those of mouse corneal cells that showed the immunological phenotype (CD103^+^) of resident memory T cells, identified in the live mice corneas after HSV keratitis using intravital TPM [28].

In that study, intra-vital TPM was used to evaluate responses of T cells in murine corneas following HSV infection. The authors found that, in response to a HSV infection, a significant number of virus-specific T cells infiltrated the cornea. Tissue-resident memory T (TRM) cells were formed as a result of CD8^+^ T cell recruitment in response to an ocular infection. These patrolling corneal TRM cells were found to be mobile and rapidly responded to antigen representation. Since prior exposure to HSV is common in humans, it may explain the prevalence of such cells in healthy individuals. Hence, the main contribution of TPM is the discovery of corneal TRM cells, which have previously been considered as dendritic cells [28].

Plasmacytoid DCs (pDCs) specialize in the detection of the nucleic acids of pathogens via the production of type 1 IFNs (IFN-α/β) in high levels, uniquely equipping them with the ability to detect viral infections such as HSV keratitis. Local depletion of pDC leads to increased severity of disease and complications such as nerve loss, dissemination to draining lymph nodes, trigeminal ganglion, and increased mortality. The local adoptive transfer of pDC conversely limits disease [29].

The technique of the multiphoton imaging of corneal explants from DPE-GFP × RAG-1−/− mice allows for the creation of a high-resolution, three-dimensional reconstruction of pDC morphology, without possible artefacts that might arise secondarily during tissue processing. Via this technique, authors observed that pDCs were limited to the anterior stroma, adjacent to the corneal epithelium, but were not found in the posterior stroma or epithelium itself. When GFP+ pDCs were compared with the native GFP+ cDCs in the CD11c-GFP = DTR mice corneas, the former was found to have a round cell body with round-ended, stub-like extensions, and a lack of dendritic processes similar to those of the latter during transmission electron microscopy. Three days after induction of sterile corneal inflammation using thermal cautery, pDC showed two different morphologies, one with elongated cell bodies and numerous thin dendritiform processes and another with rounder cell bodies without the extensions [29].

The HSV-1 strain McKrae is a neurovirulent, stromal disease-causing strain that has been triple plaque-purified. Using a 30-gauge needle, mice corneas were scarified 5 (horizontal) × 5 (vertical) times and topically inoculated with 103 or 2 × 106 PFU of HSV-1 strain McKrae in 10 μL Dulbecco’s modified Eagle’s medium (DMEM) culture media (Mediatech, Inc., Manassas, VA, USA). Mice serving as controls were subjected to the same scarification and treatment with an equivalent volume of virus-free DMEM. Multiphoton microscopy was performed on freshly excised, unstained, and unfixed corneas of DPE-GFP × RAG-1−/− or CD11c-GFP-DTR mice at an excitation wavelength of 880 nm. To assess kinetics, stacks of multiple xy sections with 3 μm z spacing were acquired every 60 s for at least 30 min to provide image volumes of at least 130 μm in depth [29].

Cell tracking experiments show that corneal pDCs move with greater mean speed during inflammation. After HSV-1 infection and after thermal cautery, median speeds of pDCs (3.4 μm/min and 3.2 μm/min, respectively) were comparable, and slightly greater after suture placement (4.2 μm/min). However, when compared to the results previously reported for cDCs in drainage lymph nodes, the speed of movement of pDCs was lower. This could possibly be attributed to the collagen fibers of the cornea being densely packed, resulting in hindered cell movements. In addition, the meandering index (a measure of directionality) remained similar after thermal cautery, HSV-1 inoculatin, and suture placement. These results suggest that corneal pDCs, while having minimal movements at baseline, display increased motility in inflammatory microenvironments [29].

In summary, advances in microscopy have led to greater understanding of anti-viral defense mechanisms in the cornea.

## 6. Ocular Surface Disease

Lining the ocular surface, airways, and gastrointestinal tracts is the mucosal epithelium, which plays a crucial role in lubricating and maintaining a barrier against pathogens and outside debris. Within the mucosal epithelium lies an abundance of goblet cells (GCs), which secrete mucins to form layers of mucus that protect the mucosal epithelium [30,31,32]. Research has shown that the delivery of antigens to dendritic cells, which produces subsequent antigen-specific T-cell responses, can occur via the formation of GC-associated antigen passages (GAPS) from intestinal and conjunctival GCs [33,34,35]. This delivery of antigens to immune cells is involved in various human diseases. In fact, dysfunctions of GAPs may be instrumental in dry eye disease and intestinal inflammation. Interestingly, GAPs were first isolated in the intestine using intravital two-photon microscopy [36].

Studies in the conjunctiva have suggested that, apart from protection against infection, pDCs play additional roles in the ocular surface. Using epi-fluorescent microscopy, pDCs in DPE-GFP × RAG1−/− transgenic mice were tagged with GFP and were found to be present in a higher density in the limbus as compared to the bulbar conjunctiva. Via intravital multiphoton microscopy, it was also found that resident pDCs follow the limbal vessel distribution and patrol the intravascular space. In humans, in vitro multiphoton microscopy demonstrated that pDCs gravitate towards and interact with umbilical vein endothelial cells during tube formation. Considering the extravascular arrangement of the pDCs adjacent to the limbal vasculature, pDCs located around the limbal vessels could have extravasated from the limbal vessels or have migrated towards the vessels from the conjunctiva, or both. In addition, pDCs may have played a role in mediating vascular development, maintenance, or repair [37]. pDCs were implicated in the pathogenesis of several autoimmune diseases, including systemic lupus erythematosus [38,39], psoriasis [40,41], and multiple sclerosis [42]. In addition, pDCs may have contributed to the progression of ophthalmic presentations in Sjögren’s syndrome [43] beyond the notation of the IFN-α signature [44]. In the same hypothetical vein, while pDCs were not found in salivary glands in disease-free states [45], they invaded these glands in patients with Sjögren’s syndrome [46].

Glaucoma is an important cause of irreversible, sight-threatening disease in the developed world. In surgeries such as glaucoma filtration surgery, success depends on the continual flow of aqueous humor of the eye from the anterior chamber to the subconjunctival space. Premature or dysregulated scarring is the main reason for the cessation of aqueous flow and consequential failure of these surgeries. Since subconjunctival scarring involves laying down abnormal and excessive amounts of fibrosis tissue, any technique that improves the visualization of the type and the amount of connective tissue in the conjunctival and subconjunctival space is highly valuable. In one study, an inverted TPM was used to image the limbal conjunctiva, Tenon’s capsule, and sclera of human donor corneal buttons. There was no need for a fixation process. For two-photon excitation, the Ti:Sapphire laser was tuned at 850 nm and second harmonic generation (SHG) and autofluorescence backscatter signals were collected via 425/30 nm and 525/45 nm emission filters, respectively. With SHG, collagen signals were obtained, while with AF, elastin signals were obtained. Consecutively, numerous and overlapping z-stack images were obtained. The collagen bundles were found to vary greatly in size and density, increasing in relation to tissue depth from conjunctiva to sclera. Collagen bundles were found to be loosely arranged and disorganized, <10 μm in width, and arranged in superficial image planes. In contrast, in the deeper image planes such as the episclera and superficial sclera, collagen bundles were densely packed, orderly, and thicker, with a width of near 100 μm. Comparatively, elastin fibers were thinner and sparse, oriented independently of collagen fibers in the superficial layers but interwoven through gaps between collagen bundles in the deep sclera. At the limbus, both collagen and elastin fibers were compact and distributed in a perpendicular manner to the limbal annulus. This study, which increased our understanding of the structure of the conjunctival and scleral matrix, sets the stage for further investigations in ocular physiology and pathology [47].

In many instances of surgical failure or conjunctival fibrosis, the cause is excessive inflammation or overly exuberant wound healing. In order to investigate these processes, a study aimed to visualize the in vivo immune cell dynamics of subconjunctival tissues’ undergoing wound healing using MPM. Mice that were gene-targeted to express enhanced green fluorescent protein under the regulation of the endogenous lysozyme M promoter (LysM-eGFP mice) were first anesthetized with isoflurane and subsequently injured via a 10-0 nylon conjunctival suture. The intravenous injection of 70 kDa rhodamine-conjugated dextran was then performed to visualize the vasculature. Three-dimensional subconjunctival tissue images were obtained every minute for 20 min before and 0.5, 3, 6, and 72 h after injury using a multiphoton microscope.

When compared to control mice without injury, an increasing number of LysM-eGFP-positive cells were found in subconjunctival tissue after conjunctival surgery in a time-dependent manner. In addition, cell velocities increased significantly until 3 h post-surgery (5.9 ± 3.2 μm/min; *p* < 0.0001) and the elevated level was sustained until 72 h after injury (5.9 ± 3.3 μm/min). The authors concluded that this type of imaging may be a useful modality for visualizing the molecular processes of wound healing after a variation of ocular trauma or injuries, such as glaucoma surgery [48].

In another study, nine New Zealand white rabbits underwent minimally invasive glaucoma surgery (MIGDS) with the implantation of PreserFlo MicroShunt. Rabbits then received subconjunctival injections of phosphate-buffered saline (PBS) or valproic acid (VPA). After 28 days, images of unstained cryosections of the conjunctival bleb were obtained using a fully automated, programmable, multiphoton imaging platform. Overt differences in the characteristics of collagen fibers were noted between the two treatment conditions. The SHG signals reaffirmed the histochemical results with high fidelity—verifying that thick collagen fibers were induced in PBS-treated tissue in contrast to the fine collagen fibers in VPA-treated tissues [49].

In summary, using TPM and MPM led to improved visualization and understanding of immunological processes and responses to injury or trauma of the ocular surface.

## 7. Stem Cell Biology

The corneal epithelial progenitor cells lie within a special anatomical compartment in the basal layer of the corneal limbus. From there, these cells are expected to undergo two kinds of movements: one centrally along the basal layer towards the center of the cornea, and another proliferating towards the superficial differentiated epithelium of the cornea [50]. Using intravital two-photon microscopy, investigators showed these dynamics for the first time in the mouse cornea [51].

In order to identify long-living lineages from the basal progenitor cells, the researchers labelled mice genetically using the *p63CreERT2* driver, together with a dual-fluorescent *ROSA26LoxP-tdTomato-STOP-LoxP-EGFP* Cre-reporter (*R26-mTmG*) and observed these mice for three months. The study team was able to acquire single-cell-resolution, consecutive full-thickness optical sections of the entire anterior eye, inclusive of the limbus. The team then reimaged the same eyes serially at different times using identical acquisition parameters. The morphology, behavior, and localization of the long-living clones emerging from the niche were found to be heterogenous. They found, as expected, larger cell lineages that exited the niche from the inner limbus and expanded centripetally toward the central cornea. These cells replaced the smaller, short-lived, randomly placed cells in the central cornea [51].

But there were also discrete smaller outer limbal clones that expanded solely within their original location and never left the niche. These stationary limbal clones were located at and organized parallel to the limbal stromal blood vessels and collagen fibers, which were circumferentially aligned. With the use of further in vivo reporters, corneal nerves and certain immune cells were also found in this compartment of the limbus. The team found that the outer limbal stem cells were not involved in the normal corneal homeostasis, and that only after extensive epithelial injury such as in a scrape wound of the cornea would the outer limbal clones exit their compartment to migrate centrally [51].

Next, the investigators used the *p63CreERT2* driver to first induce the expression of the H2B-GFP fusion reporter in all epithelial basal cells. They then established that the expression of the reporters were at equivalent intensity in basal cells across all epithelial compartments. Subsequently, doxycycline was added to mice diet to reduce the expression of H2B-GFP. The gradual reduction in the H2B-GFP signal was noted during the actively dividing and cycling epithelial cells in the subsequent chase period. At 1 month, the sole remnant cells that divided the least remained visible. By quantifying cell fate (cell division time and terminal differentiation events), the authors identified four types of stem cell behavior. First, the stem cells in the outer limbus showed self-renewal, slow cycling, and limited growth potential. Second, the stem cells in the inner limbus showed mostly symmetric cell divisions and exhibited the greatest potential for long-term growth. Among the transient cells, the ones in the peripheral cornea showed a higher tendency to divide symmetrically and expand, while those in the central cornea were very short-lived and rapidly terminally differentiated [51].

The authors performed a three-dimensional analysis of the transit trajectories of differentiating cells. They identified the transit time of the basal cells to the most superficial layer of the cornea and found this to be 4–5 days. In the limbus, as there were fewer suprabasal layers, the effective speed of stratification was lower than in the rest of the cornea. The terminally differentiated cells in the cornea interestingly followed an arced path in a centrifugal direction through suprabasal layers, while those at the limbus transited in a direct upward trajectory [51].

In contrast to labeled cells in the limbus, conjunctival progenitors marked in the above fashion showed no notable polarization in their growth patterns over time.

The authors wondered what stopped the inner limbal clones from transiting both laterally and in the conjunctival direction. They mentioned that outer limbal cells may be essential in keeping a barrier between the cornea and conjunctiva. Mechanical forces and adhesion between cells may contribute to the unusual observed movement of suprabasal cells [51].

Undoubtedly, TPM contributed to our understanding of the ocular stem cell niche and will continue to play a major role in the era of stem cell therapy for ocular surface disorders.

## 8. Ocular Drug Delivery

Chronic fungal infections of the cornea are a significant cause of blindness worldwide. In fungal keratitis, fortified amphotericin B (AMP-B) eyedrops are the first-line therapy of choice [52]. However, the poor solubility and penetration of this drug through intact cornea leads to poor bioavailability, and lack of efficacy [53]. Microneedles such as the biodegradable polyvinylpyrrolidone (PVP) and hyaluronic acid (HA)-based devices are minimally invasive and may be used to overcome the drawbacks of amphotericin. The biodegradable polymers reduce drug toxicity and prolong the drug retention time. As part of the in vitro studies using porcine eyeballs, sclerocorneal tissue was treated with Amphotericin loaded microneedles, and multiphoton microscopic studies were performed to study the depth of AMP-B penetration and distribution within the corneal tissues. In the following five minutes after the above treatment, via the use of 3D imaging for AMP-B MN arrays, traces of AMP-B were seen on the tissue’s surface and within the corneal tissue at the depth of 0.4 mm in the Bowman’s layer, confirming the penetration of microneedles compared to the control with amphotericin B solution [54].

## 9. Limitations and Future Directions

Future directions in the use of intravital two-photon microscopy will depend on the development of reporter mice with improved labeling with respect to the specificity of cells and tissues [55]. This is evident in the discussion of corneal immunology above. A limitation of using mice to understand mechanisms in these experiments is that there may be significant differences in the immune system and corneal physiology between the mouse and human.

To address this limitation, we foresee the use of multi-photon microscopy in vitro, using sections from humans, to increase in the future, whether in the area of wound healing, viral infection, or drug delivery.

However, limitations of TPM and MPM use in humans include suboptimal image quality due to poor image stability. Methods to circumvent this include using a chin rest similar to that of the slit lamp, where the patient’s head is accurately positioned in relation to the control cameras and laser machine and subsequently immobilized, and the use of a fixation target to prevent macromovements. However, uncontrolled micromovement of the eye has still been found to compromise image quality, especially so in longer imaging timings. Hence, the utilization of fast scanning protocols involving the inculcation of safety limits regarding pixel dwell time, pixel-based image dimensions, and image region set would also optimize quality of images obtained [56].

In general, especially in the absence of transgenic mice that express fluorescent molecules in specific tissues, the specific labeling of cells in vivo with exogenous probes is challenging. Techniques like in vivo transfection require special methods to ensure the stability and efficient expression of the gene.

Other limitations of two-photon and MPM imaging include issues related to image sensors, fluorescence decay, and scenarios with low signal-to-noise ratio (SNR). The attenuation of signal with depth also results from scattering and absorbance of the laser. This attenuation is reduced by the use of near-infrared light that falls in the range of the “optical window” of tissues to minimize absorbance and scattering. Rapid image collection is necessary for many forms of in vivo microscopy as this minimizes the effects of movement that can seriously compromise image quality. Even though animals may be anesthetized, there would still be movements related to breathing, etc. High-speed laser scanning microscopy can be accomplished by “parallelizing” the illumination, so that multiple points of illumination simultaneously stimulate fluorescence from multiple points in the sample. This “tandem scanning” increases the speed of the microscopy in proportion to the number of illumination foci. However, tandem scanning microscope systems are generally unsuitable for thick tissues, as adjacent beams interact with each other away from the focus, resulting in increase in background fluorescence. Because of the high magnification of the imaging and the focus on a small area of the ocular tissue, depending on the presence or absence of neighboring landmarks, it may be difficult to localize subsequent imaging to exactly the same area for longitudinal imaging. In contrast to TPM, which measures the intensity of fluorescence emission, fluorescence lifetime imaging microscopy (FLIM) detects tissue pathology by monitoring the fine-grained (~nanosecond-scale) temporal decay of fluorescence emission. FLIM, while postulated to have advantages in distinguishing fluorophore types and changes in an environment better than TPM [57], does however require the capture of photon timing information with sub-nanosecond precision. If there are very few photons (photon-starved regime), SPC pixels suffer from unreliable estimates due to poor SNRs [58]. If there are too many photons (photon-flooded regime), the measured photon transients suffer from severe non-linear distortions called pileup. Investigators have proposed guided photon processing to recover the photon fluxes in low-SNR scenarios. Guided photon processing uses spatial frequency correlations between the intensity image and the photon transient cube [59].

## 10. Conclusions

TPM and MPM have contributed significantly to scientific progress and clinical translation in diverse fields ranging from viral immunology and stem cell biology to drug delivery. With the increased availability of such commercial imaging systems, more laboratories will be employing these techniques. More exciting advances can be expected in the near future, with the development of improved reporter mice as well as improvements in optics and imaging.

## Figures and Tables

**Figure 1 ijms-25-01670-f001:**
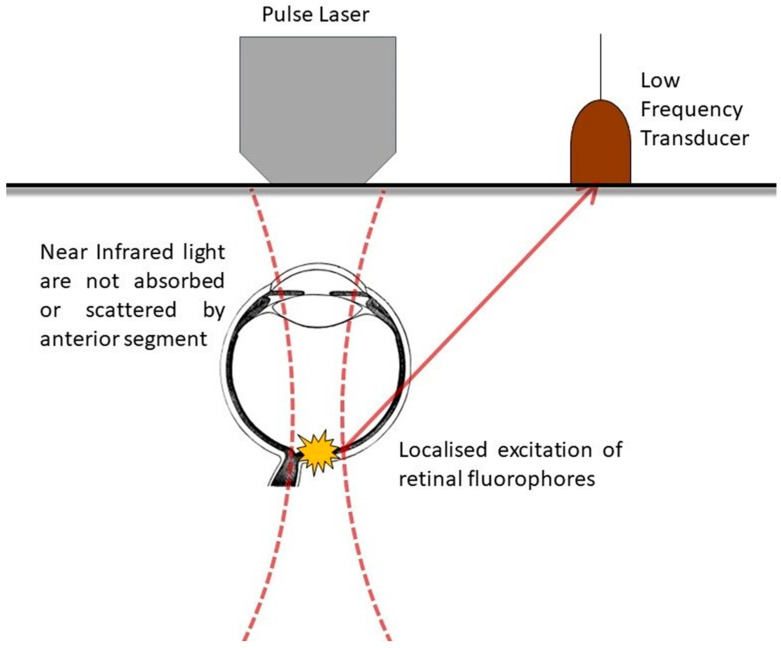
Schematic describing the use of TPM in retina imaging. Near infrared light is emitted by the pulse laser and induces localized excitation of retinal fluorophores, with their subsequent generated fluorescence detected by the low frequency transducer.

**Figure 2 ijms-25-01670-f002:**
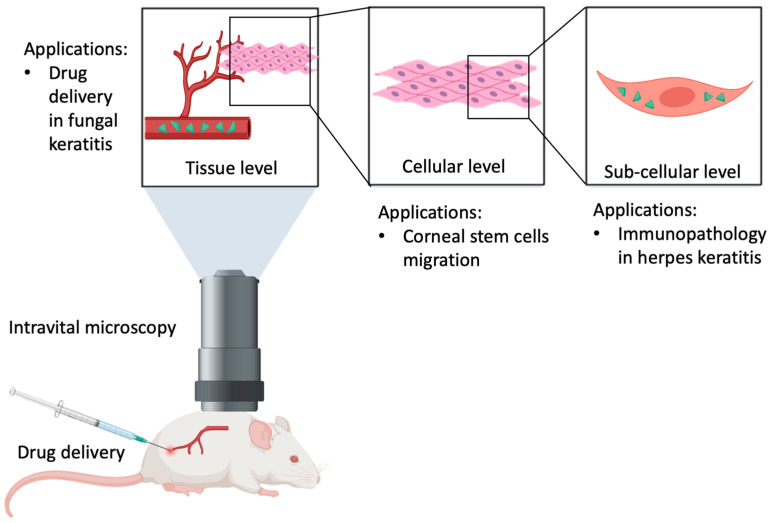
Application of two-photon microscopy to various ocular tissues, physiological and pathological processes.

## Data Availability

Not applicable.

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
