# Peer review of "Two-Photon and Multiphoton Microscopy in Anterior Segment Diseases of the Eye"

_ijms, 2024, doi:10.3390/ijms25031670_

Round 1
Reviewer 1 Report
Comments and Suggestions for Authors
-
Comparison with Baseline Modalities: The manuscript would benefit from a more detailed comparison of Two-Photon Microscopy (TPM) with baseline imaging modalities like Optical Coherence Tomography (OCT) in retinal imaging. This comparison should highlight the advantages and limitations of TPM in a comprehensive manner.
-
Safety and Effectiveness in Clinical Use: It's crucial to address the maximum safe laser intensity for TPM in clinical settings, particularly in eye imaging. The manuscript should also discuss the imaging speed in relation to patient comfort, potential issues like fluorescence quenching, its impact on imaging results, and strategies to mitigate such effects. Furthermore, elaboration on the imaging depth and depth resolution of 3D imaging.
-
Improvements in Figure 1: Figure 1 needs enhancement for clarity and accuracy. It should depict the illumination and detection paths, and detail the objectives and detectors used. A representative retinal image should be included as well as a detailed description of the imaging pipeline from preparation to post-processing.
-
Enhancing Figure 2: This figure should offer more than just structural information from tissue to subcellular. It should have applications at each level of observation and discuss the pros and cons at each level.
-
Inclusion of More Figures: Each application section in the manuscript should be supported with additional figures to better illustrate the concepts, particularly highlighting the method's strengths and weaknesses in each application.
-
Comparative Analysis with Other Imaging Techniques: A comparison of TPM with other imaging modalities like OCT and Fluorescence Lifetime Imaging Microscopy (FLIM) would offer readers a clearer understanding of TPM's capabilities. Particularly, as FLIM is based on fluorescence lifetime, not intensity, it can give a better quantitative measurement then intensity based method. Can the author discuss the potential of FLIM in retinal imaging? More interestingly, the author can consider 3D imaging. Below are some papers that explains 3D FLIM, uses FLIM in retinal imaging and other clinical applications, as a reference
[1] Dysli, Chantal, et al. "Fluorescence lifetime imaging ophthalmoscopy." Progress in retinal and eye research 60 (2017): 120-143.
[2] Ma, Yayao, et al. "Light-field tomographic fluorescence lifetime imaging microscopy." Research Square (2023).
-
Data Processing Techniques: The author should discuss data processing methods like averaging and filtering, emphasizing the importance of imaging processing techniques in TPM for data interpretation and understanding.
Reviewer 2 Report
Comments and Suggestions for Authors
Review of the material entitled ‘Two-photon and Multiphoton microscopy in anterior segment 2 diseases of the eye’
The aim of the review is to describe the contribution of two-photon microscopy to ocular surface and anterior segment research.
The review must include a brief summary of the process of two photon microscopy. It must also include three photon microscopy and explains why authors have chosen only two photon.
- ‘Condensation products such as A2E’ A2E must be explain.
- The term MPM must be explicitly explain. What does it mean? Does it concern only two photon fluorescence or other processes?
- Authors say that ‘In summary, we expect TPM and MPM to play major roles in the understanding of ocular surface physiology and histopathology.’
What do you expect to see?
- ‘To address this limitation, we foresee the use of multi-photon microscopy in vitro, using sections from humans, to increase in the future, whether in the area of wound healing, viral infection, or drug delivery.’
This part must be developed.
- FLIM is not explained.
- A scheme must be given explaining the set-up of TPM and MPM.
- A tab summing up the current techniques for observations must be given with references.
Round 2
Reviewer 1 Report
Comments and Suggestions for Authors
concerns are addressed.